# A Deep Learning Approach for Mental Fatigue State Assessment

**DOI:** 10.3390/s25020555

**Published:** 2025-01-19

**Authors:** Jiaxing Fan, Lin Dong, Gang Sun, Zhize Zhou

**Affiliations:** 1Institute of Artificial Intelligence in Sports, Capital University of Physical Education and Sports, Beijing 100191, China; 21008001049@cupes.edu.cn (J.F.); donglin@cupes.edu.cn (L.D.); sungang@cupes.edu.cn (G.S.); 2Emerging Interdisciplinary Platform for Medicine and Engineering in Sports (EIPMES), Beijing 100191, China

**Keywords:** mental fatigue, deep neural network, electrocardiogram, ECG

## Abstract

This study investigates mental fatigue in sports activities by leveraging deep learning techniques, deviating from the conventional use of heart rate variability (HRV) feature analysis found in previous research. The study utilizes a hybrid deep neural network model, which integrates Residual Networks (ResNet) and Bidirectional Long Short-Term Memory (Bi-LSTM) for feature extraction, and a transformer for feature fusion. The model achieves an impressive accuracy of 95.29% in identifying fatigue from original ECG data, 2D spectral characteristics and physiological information of subjects. In comparison to traditional methods, such as Support Vector Machines (SVMs) and Random Forests (RFs), as well as other deep learning methods, including Convolutional Neural Networks (CNNs) and Long Short-Term Memory (LSTM), the proposed approach demonstrates significantly improved experimental outcomes. Overall, this study offers a promising solution for accurately recognizing fatigue through the analysis of physiological signals, with potential applications in sports and physical fitness training contexts.

## 1. Introduction

As sports participation rises, so does the likelihood of injuries [1]. These injuries often stem from improper posture, training methods, and fatigue [2,3]. Among these factors, fatigue plays an important role in affecting people’s cognitive ability and endurance performance. Fatigue, characterized by tiredness or diminished energy [4], results from various factors like physical and mental states [5], environmental conditions, individual differences [6], etc. To prevent over-fatigue and related injuries, athletes and coaches should carefully arrange training plans and exercise intensity, while also enhancing guidance and training on sports postures and techniques.

Fatigue can be broadly categorized into two types: exercise fatigue, also referred to as physiological fatigue, and mental fatigue, also known as psychological fatigue [7]. Exercise fatigue usually occurs in the muscles utilized during exercise and is characterized by a decrease in muscle strength, ultimately leading to reduced exercise capacity. Mental fatigue, on the other hand, arises in the central nervous system that controls body movements and is generally caused by demanding, prolonged, or repetitive cognitive tasks. Although the exact mechanism of mental fatigue and the role of neural mechanisms remain unclear [2,8], it is generally believed to manifest as a state change in psychobiology. That is, when an individual engages in repetitive or prolonged activities, the function of some central nervous system cells is inhibited due to sustained mental tension [4].

Mental fatigue is closely linked to various aspects such as cognitive function, emotional state, biochemical level, and neuroplasticity, potentially adversely affecting an individual’s social interactions and interpersonal abilities. It may result in issues like distraction, reduced alertness, and diminished situational awareness, exacerbate work stress and psychological burden, and ultimately impact physical health [9]. Furthermore, mental fatigue can lead to kinematic changes such as movement deformation or shifts in the center of gravity, increasing the risk of injury [1,10]. Additionally, mental fatigue and exercise fatigue are closely related [11]. For instance, mental fatigue may cause a decrease in endurance [12,13], thereby influencing sports performance [14,15].

Assessing mental fatigue before sports activities is crucial for the performance of athletes [4,14,15]. As fatigue increases, athletes’ endurance gradually decreases, and their movements will undergo kinematic changes, increasing the risk of injury [1,10]. Addressing mental fatigue promptly by adjusting training rhythm and exercise intensity will ultimately enhance training efficiency and benefits. However, unlike exercise fatigue, mental fatigue is challenging to identify through direct observation. There are primarily two methods to assess mental fatigue: subjective questionnaires, and physiological and biochemical index detection. Subjective questionnaires, such as Karolinska Sleepiness Scale (KSS), Stanford Sleepiness Scale (SSS), Visual Analogue Scale (VAS), and Chalder Fatigue Scale (CFS), face limitations due to varying participant understanding of fatigue, medical knowledge, research purposes, and scale uniformity [16]. Physiological and biochemical indicators offer higher objectivity and reliability with precise experimental data. Biochemical indicators, obtained through blood and body fluid testing, are more objective than physiological signals but require longer experimental cycle and complex steps. Physiological indicators, derived from analyzing signals such as brain or cardiac electrical signals, are readily available, non-invasive, and real-time, making them more practical for daily use.

This study focuses on mental fatigue assessment using physiological signals based on artificial intelligence methods, primarily employing psychological paradigms to intervene the mental fatigue states of subjects. These psychological paradigms include various forms such as auditory tasks, tactile tasks, and, most frequently used, visual tasks. During the completion of visual tasks, mental fatigue is induced by maintaining subjects’ high mental consumption [17]. Common visual tasks include the N-back task [18], Serial-7 Subtraction Arithmetic Tasks [19], Wisconsin Card Sorting Test [20], Forward Digit Span Tasks [21], Stroop paradigm [22,23], etc. Among them, the N-back and Stroop tasks are frequently utilized to induce mental fatigue. However, participants may rapidly grasp the N-back task’s rule, resulting in insufficient mental expenditure [24]. In contrast, the Stroop paradigm, based on a color naming experiment with distractors, has stronger randomness. The classic Stroop task slows responses of subjects due to additional cognitive load when colors are inconsistent with letters. Stroop’s effectiveness in inducing mental fatigue is demonstrated by long-term experiments [25] and the change trend in the inverse efficiency score (IES) of the 60-minute Stroop task [26]. Stroop has also been used as a control condition, affirming its validity [27]. In this study, the Stroop paradigm is adopted to induce mental fatigue, as it exercises attention and executive control abilities simultaneously, and allows for adjustable setup and difficulty according to the actual situation.

Physiological signals for fatigue detection are then acquired through specialized equipment. Common physiological signals used in fatigue detection include electroencephalogram (EEG) signals [8,28,29], electrocardiogram (ECG or EKG) signals [30,31], electromyography (EMG) signals [32,33], photoplethysmography (PPG) signals [34,35], and more. In this paper, fatigue analysis focuses on ECG signals, which reflect cardiac rhythm, known to be closely related to the activity state of the autonomic nervous system [36]. Moreover, a low-cost portable device is employed to collect ECG signals. Compared to other equipment types, portable ECG data collection devices offer advantages of convenience and ease of use, enabling real-time monitoring and data analysis while on the move. This facilitates individuals to better manage their fatigue status, thereby maintaining good working and living conditions.

The collected ECG signals undergo preprocessing steps, such as noise reduction and standardization, to enhance the signal quality and improve the accuracy of the analysis results [37]. Early studies focused on the quasi-periodicity of the signal to extract relevant features from the preprocessed ECG signal as the analysis object. The most critical feature is heart rate variability (HRV), which can usually be calculated by determining the dispersion and mean of the heartbeat time. Feature extraction methods are generally divided into two categories: manual feature extraction and automatic detection [38]. Traditional statistical methods are typically used to evaluate the extracted features [39]. In recent years, machine learning methods, such as Decision Tree (DT) and Random Forest (RF), have also been applied in mental fatigue assessment [40]. Additionally, Principal Component Analysis (PCA) has been used to perform daily fatigue classification tasks [21]. However, due to the tedious feature extraction steps and the lack of comprehensive information, the accuracy of feature-based fatigue analysis is usually insufficient.

With the development of deep learning, existing research for ECG signal analysis has gradually shifted to using deep neural networks (DNNs) [41,42,43], especially in heart disease [44]. Deep Belief Net (DBN) exhibits certain advantages in identifying atrial fibrillation [45]; however, it requires a considerable amount of time to process long sequence data. As a representative of timing analysis algorithms, Long Short-Term Memory (LSTM) outperforms DBN in distinguishing ECG morphological differences and understanding context [46]. Then, a DNN is proposed for automatically extracting features of ECG signals [41], allowing for parallel processing of multiple signal samples and enhancing processing efficiency. Furthermore, Global Regression Neural Network (GRNN) improves the model’s classification performance in heart disease by boosting its generalization ability [47]. Simultaneously, research on mental fatigue has begun to incorporate deep learning methods as well. Deep Sparse Auto-encoding Network (DSAEN) exhibits advantages in processing incomplete and inaccurate signal data by combining with a SoftMax classifier [48]. A deep spatiotemporal convolutional bidirectional LSTM network is employed to analyze Force Feedback Glove (FFG) signals in the study of pilot’s mental fatigue, achieving an accuracy of 87% on the fatigue classification task [49]. Comparatively, deep learning models typically analyze the complete original signal directly, reducing reliance on manual feature extraction for processing complex data features and mitigating errors caused by human factors [44,50], thereby being more robust to anomalies in the data.

By incorporating time–frequency domain features extracted from ECG signals into DNNs, it is believed that the utilization of signal features can be optimized. Time frequency domain conversion of ECG signals usually use short-time Fourier transform (STFT) or wavelet transform to create a two-dimensional (2D) spectrogram. In heart-disease-related fields, 2D spectrograms are used as input for U-Net, enhancing model accuracy by incorporating both global and local information compared to simple 1D time series data features [44]. Two-dimensional spectrograms are also used in Convolutional Neural Networks (CNNs) with a parameter sharing mechanism that reduces the number of network parameters and model complexity [51]. Additionally, 2D spectrograms are used in a transformer network that integrates the features of the Multi-head Self-attention mechanism and Position Encoding, improving model understanding and classification accuracy of complex ECG patterns, as well as training speed and efficiency [52]. To the best of our knowledge, existing research has primarily focused on applications in heart disease, with few studies exploring the use of ECG spectrograms for mental fatigue classification.

In this paper, a method is proposed with a hybrid DNN model for mental fatigue assessment. The Stroop experiment is initially applied to induce mental fatigue in subjects, and the corresponding ECG signals are collected using portable devices. Subsequently, the ECG signal data are preprocessed and segmented into 1D time series. These time series are then converted into spectrograms in the time–frequency domain. The time series features of ECG and STFT spectrograms features are fed into feature extraction networks comprising a Residual Network (ResNet) and Bi-directional Long Short-Term Memory (BiLSTM). ResNet enhances performance and trainability by incorporating residual connections, effectively capturing ECG signal features. BiLSTM further augments expressive capabilities by processing the time series bidirectionally, thus capturing periodic patterns, long-term dependencies, and complex time dynamics in ECG signals. Moreover, to account for baseline differences in ECG signals among different subjects [53], individual basic physiological information, including age and gender, is incorporated into the hybrid model to improve classification accuracy. Specifically, physiological information and the extracted time series and spectrograms features are integrated through a transformer, which uses the attention mechanism for better contextual information capturing. In the experiment, a 130 Hz dual-lead portable ECG acquisition device (Polar H10, Polar Electro Oy, Finland) is utilized to achieve low-cost and high-precision mental fatigue assessment, and the iOS app "ECG EKG Recorder for Polar H10" was used (provided by developer Philipp Poeml, compatible with iOS 12.0 and later versions). The hybrid model was compared with RF, Support Vector Machine (SVM), LSTM, and CNNs, achieving an accuracy of 95.29%, which outperformed the other methods.

## 2. Methods

The proposed mental fatigue assessment method is outlined in Figure 1. The process begins with a 60-minute Stroop experiment (Figure 1b) conducted on the subjects (Figure 1a) to induce mental fatigue. The ECG signals are recorded using portable wearable heart rate belts and a third-party APP, and subjects are asked to mark the score of the Visual Analogue Scale (VAS) to evaluate their mental fatigue states (Figure 1c). The ECG signals are then denoised and standardized to enhance data quality and accuracy (Figure 1d). Following noise reduction, the ECG signals are segmented into 1D heartbeat time series (Figure 1e), which are subsequently transformed into 2D spectrograms using STFT (Figure 1f). The 1D time series and 2D spectrograms, combined with subjects’ basic physiological information, are fed into the proposed hybrid DNN to infer the mental fatigue state, which consists of the following two steps: feature extraction (Figure 1g) and feature fusion (Figure 1h).

### 2.1. Experimental Setup

Prior to the experiment, participants are instructed to wear Polar H10 heart rate belts on their chests, which measure electrical signals generated by the heart’s contraction and relaxation, as reflected in chest pressure changes. The belts are set to engineering debug mode in order to collect original ECG signals, sampled at 130 Hz. The date is then recorded using ECG Recorder, a third-party IOS application, with the values saved in files named by timestamps on a storage device.

Subsequently, the participants are required to complete the Stroop task to induce mental fatigue. As illustrated in Figure 2, the conventional Stroop task involves disregarding the word’s meaning and focusing solely on the ink color. In our experiment, the task is modified to increase its difficulty. Participants are not only required to identify the ink color by pressing the first letter of that color on the keyboard, but also to recognize the word’s meaning by vocalizing it in Chinese simultaneously. For example, if “green” with a red ink color is displayed, participants need to say “green” in Chinese and press the letter “R” concurrently. This lengthy and taxing mental exercise aims to elicit fatigue, as evidenced by a delayed reaction time when the word’s meaning diverges from the ink color, referring to the Stroop effect [51,54].

During the task interval, participants were asked to complete the VAS [55] to assess their mental fatigue levels. VAS is the most widely used tool for assessing mental fatigue [56]. Despite being a subjective self-reporting method, its validity and reliability have been well-documented in prior research [56]. The VAS employed in this study consists of an unlabeled line segment with a length of 10 cm. Additionally, the left end of the line segment symbolizes a state of absolute non-fatigue, whereas the right end indicates complete fatigue. Subjects were asked to mark their current level of fatigue at any point along the line segment. Subsequently, the ECG signals were labelled as “fatigue” (F) if the VAS score was above 80%, and as “non-fatigue” (N) otherwise.

### 2.2. Data Preprocessing

The original ECG signals are then denoised through decomposition and reconstruction using wavelet transformation, specifically the Daubechies-5 (DB-5) wavelet [57,58]. The denoising process begins by decomposing the signals into distinct frequency bands to extract features at various scales. This decomposition consists of an approximation coefficient (cA9) and eight detail coefficients (cD9, cD8,…, cD2, cD1). Then, the coefficients are subjected to predefined thresholds for denoising and utilized to reconstruct the denoised signal data through wavelet inversion. The denoised signals are further standardized by dividing each feature dimension by its mean and variance. The combination of denoising and standardization guarantees that the input data are of high quality and well-prepared for the neural network [37,59].

The filtered data are segmented into time series, as shown in Figure 3a,b, each comprising approximately 1 s of data at a sampling rate of 130 Hz. To be more specific, segmentation occurs once every 130 sampling points. Furthermore, these time series are organized into groups as shown in Figure 3b, with each group containing 10 consecutive time series. This grouping strategy enables the aggregation of 10 s of data into a single set of features, facilitating the subsequent process of feature extraction and analysis.

After signal segmentation, STFT is applied on each time series to obtain the corresponding spectral information, as shown in Figure 3b,c. STFT shifts a window with function $g\left(n\right)$ over a time series by a fixed step and performs the Fourier transform on that window, as described by the following equation:(1)$$S\left(n,k\right)=\sum_{m=0}^{N-1}x\left(m\right)g\left(n-m\right)e^{-j\frac{2\pi km}{N}}$$
where $S\left(n,k\right)$ represents the time–frequency element at time n and frequency $k,\;x\left(n\right)$ denotes point n of the signal in the time domain, N is the number of samples in each window, and $e^{-j\frac{2\pi km}{N}}$ is the complex exponential term used for computing the Fourier transform.

By shifting the window in time, STFT captures the spectral features of the signal in a time–frequency representation, rendering it highly suitable for processing nonstationary signals with changing spectra over time. Although STFT sacrifices some time and spectral resolution, it still provides valuable information about the signal’s time–frequency distribution.

### 2.3. Deep Neural Network

A hybrid DNN is used for final mental fatigue assessment, which is comprised of two steps: feature extraction and feature fusion. Feature extraction involves extracting features from time series, STFT spectrograms, and physiological information. Feature fusion uses a transformer model to integrate these features for classifying mental fatigue state. A cross-entropy loss function guides the network to optimize predictions by minimizing the loss.

**Feature extraction**: The feature extraction process is illustrated in Figure 4. Each time series and its corresponding 2D spectrogram are fed into two different models with the same structure, specifically, an integration of a Residual Network (ResNet) and bidirectional LSTM (BiLSTM), as shown in Figure 4a,b. ResNet consists of multiple residual blocks, each containing two convolutional layers for feature learning and dimension adjustment. Additionally, the convolutional layers of ResNet are 1D layers for time series data and 2D layers for the image input, as shown in Figure 4a. By merging input features with learned residuals, ResNet ensures effective information transmission between layers, capturing contextual cues to enhance feature detection and representation capabilities. Additionally, ELU is used as the activation function, addressing gradient disappearance and fitting issues. BiLSTM is an extension of LSTM that comprehensively considers context information, synthesizing information from both forward and backward directions to better understand the sequence data and extract stronger feature representations. In the BiLSTM network, each layer has hidden states and memory units that capture contextual information about the sequence data, effectively capturing temporal dependencies and performing iterative calculations for time steps.

The features of physiological information, comprising the subjects’ age and gender, are effectively captured using one-hot encoding techniques, as shown in Figure 4c. A 100-dimensional vector is constructed for age encoding, with one dimension representing the age and the rest assigning zero values. Word2vec encoding is utilized to transform the corresponding gender-related word into a lower-dimensional vector. Subsequently, both age and gender vectors are fed into a 6-layer fully connected neural network to further extract significant features from the encoded data.

**Feature fusion**: The three extraction features are fed into a transformer encoder to effectively blend them together, as shown in Figure 5. In a basic transformer layer, the multi-head self-attention mechanism plays a crucial role by calculating weights to ascertain the importance of different inputs and capture their meaningful dependencies. The fused features then undergo further transformation through a feed forward module. After passing through multiple layers of the transformer encoder, the features are ultimately linearly transformed and mapped by a fully connected layer to obtain the mental fatigue state. This fusion of diverse features enables a comprehensive understanding of the ECG signal data, images features, and physiological information, leading to more accurate fatigue detection.

**Loss function**: The cross-entropy loss function is employed in this hybrid model, since it allows the model to learn parameters with good classification ability and improve the performance of classification tasks for the training of neural networks. For a given classification problem with N samples and g categories (g=2 in our case), the cross-entropy loss can be expressed as follows:(2)$$L=-\frac{1}{N}\sum_{q=1}^{N}\sum_{r=1}^{g}S_{qr}\log\left(P_{qr}\right)$$
where $P_{qr}$ represents the predicted probability for sample q belonging to category r, and $S_{qr}$ represents the label of ground truth; specifically, $S_{qr}$ is 1 if the sample belongs to category r and 0 otherwise.

## 3. Results

### 3.1. Preparation

To conduct the experiment, a total of 42 participants were recruited, comprising 28 males and 14 females. The participants were all students from Capital University of Physical Education and Sports, aged between 18 and 27. As the university is focused on sports, most of them are professional or semi-professional athletes. All participants fulfilled the experimental criteria, including having sound mental health, absence of visual or cognitive impairments, and no history of cardiovascular disease. To ensure the accuracy of the data, it is imperative to adhere to certain conditions prior to conducting the experiment, such as receiving adequate and regular sleep, avoiding caffeine, and adhering to any prescribed cardiovascular treatments. Additionally, participants were instructed to refrain from engaging in strenuous mental and physical activities, maintain a regular diet, and ensure adequate sleep the day before the experiment. This study has obtained ethical approval from the ethics committee and has been conducted in accordance with the Declaration of Helsinki.

### 3.2. Dataset

To validate our method, a dataset was created as, to our knowledge, no public physiological data on mental fatigue are available. Participants began by completing a registration form to provide essential physiological information. Next, they were instructed to wear Polar H10 heart rate belts for ECG signal recording. Subsequently, they were asked to fill out the initial VAS score, which helps objectively assess participants’ perception of their condition at the start of the experiment. Concurrently, pre-ECG data collection was conducted to record their resting heart rate for five minutes.

Having completed the above steps, participants were asked to perform the Stroop task using a computer program. They were given one minute to understand the instructions and familiarize themselves with the task flow. Participants then completed six rounds of the Stroop task, each lasting ten minutes, with a five-minute break between rounds. During the break, ECG data were collected, and participants were required to complete the VAS score to indicate their fatigue or energy levels. To preserve data accuracy, major anomalies discovered prior to data collection were eliminated. The experiment ended if the VAS reached 80% of the upper limit after the sixth round; otherwise, it continued until the VAS reached the 80% threshold.

The collected ECG signals were denoised, as illustrated in Figure 6a,b. The Dobesi wavelet-5 segmentation algorithm was adopted for denoising due to its effectiveness in eliminating noise while retaining important signal information. Additionally, the DC component in the signal was removed. Subsequently, the signals were standardized, as shown in Figure 6b,c. Each signal was first centered by subtracting the average value of all the points and then divided by its standard deviation.

Then, the preprocessed data were segmented into 1D time series and subsequently converted into 2D spectrograms using STFT, as shown in Figure 7. In the meantime, the label of mental fatigue (F) or non-fatigue (N) was assigned to the collected data based on the VAS score. The dataset was divided into a training set and a test set at a ratio of 9:1 for evaluating the performance and generalization ability of our method on unseen data. The accuracy and F1 score are employed as experimental evaluation metrics.

### 3.3. Quantitative Comparison

Quantitative experiments are conducted to validate our method, and the results are compared with commonly used models, including SVM, RF, LSTM, and CNNs, as shown in Table 1.

SVM is a supervised learning technique commonly used to differentiate datasets that are either linearly separable or roughly separable using support vectors as decision boundaries. In this study, the SVM method utilizes a regularization parameter (penalty term) of 0.1 to optimize the balance between fitting and generalization ability. The gamma parameter that controls the influence range of the kernel function is set to 10, and a sigmoid kernel function type is selected. Proper gamma adjustment is crucial for handling nonlinear problems to avoid over- or underfitting. The results in Table 1 indicate that SVM performs poorly on this dataset, with a low F1 score (0.38) and accuracy (55.11%). This could be due to SVM’s limitations in modeling complex nonlinear problems, since the ECG signal data in this study exhibit complex time dynamics and nonlinear characteristics.

RF, an ensemble learning algorithm composed of multiple decision trees, improves model accuracy and generalization through collective decision-making. Random sampling is used to create different training sets for building independent trees in RF. In this study, 100 decision trees are used for reproducible results, with a random seed value of 40. The maximum depth of each decision tree is set to 100. RF tends to be more tolerant towards outliers and missing data, but the decision-tree-based approach may not effectively capture hidden dynamic features and complex nonlinear relationships in the ECG signal time series. Thus, the F1 score and accuracy for RF are only 0.57 and 62.26%.

CNNs represent another widely adopted approach, and in this study, the CNN algorithm is used with STFT spectrograms for feature extraction. The convolution layer applies convolution to the input data, extracting features, while the pooling layer reduces the feature map size through downsampling. An 18-layer Convolutional Neural Network is utilized, with a convolutional kernel size of 3 × 3. CNNs show better performance in this study, with an F1 score of 0.54 and an accuracy of 69.02%. Compared to SVM and RF, CNNs outperform in terms of performance. The combination of convolution and pooling layers allows for effective extraction of local and global features from the input, enhancing the model’s representation ability for ECG data. CNNs also possess the advantage of automatically learning feature representations without manual extraction, improving the accuracy of ECG signal data representation.

LSTM networks are effective in capturing and preserving temporal information, particularly with long sequences. By employing memory units and gate mechanisms, they adeptly retain context information within sequence data. This ability to encode and preserve temporal details allows LSTM to excel in timing modeling and sequential data processing, resulting in improved performance. In this study, LSTM has a depth of 6 and an input/hidden state dimension of 64. The introduction of LSTM significantly enhances experimental performance, with an accuracy of 86.67% and an F1 score of 0.87. Furthermore, Bi-LSTM, with its dual-directional processing mechanism, captures even richer contextual information compared to standard LSTM. In our experiments, Bi-LSTM demonstrates superior performance, achieving an accuracy of 90.20% and F1 of 0.90.

While each algorithm’s performance progressively improves from SVM to RF, CNNs, and LSTM, the hybrid algorithm proposed in this study achieves an even higher accuracy of 95.29% and an F1 score of 0.95. This demonstrates that the experimental approach used in this study can classify data more accurately. The hybrid algorithm incorporates a combination of deep neural networks and transformers, offering advantages over the aforementioned four algorithms in terms of performance.

### 3.4. Ablation Study

An ablation study is further conducted to demonstrate the rationale behind the adoption of the hybrid algorithm. The experimental results are shown in Table 2. When the spectrogram extracted features were used alone for classification tasks, the F1 score was 0.69 and the accuracy was 69.41%. When the extracted spectrogram features and physiological information features were used as the final input of the model at the same time, the F1 was 0.71 and the accuracy was 71.37%. When ECG time series data were used as features, the F1 and accuracy were 0.89 and 89.41%, respectively. ECG time series data and spectrum feature fusion resulted in an F1 score of 0.92 and accuracy of 92.54%. Combination of ECG time series data features and physiological information features achieved an F1 score and accuracy of 0.93 and 93.33%, respectively. When all three features were used together, the F1 score increased to 0.95 and the accuracy increased to 95.29%.

Based on the experimental results, it has been demonstrated that the mixed scheme of S and P is superior to using S and T networks individually. The mixed scheme has shown an improvement in classification accuracy of 3.92% and 1.96%, respectively. By combining the two feature data, a more accurate representation of features is achieved, enabling the model to better understand the correlation between ECG signal data and physiological feature information. As a result, the classification performance and robustness of the model are enhanced.

Using a single feature (T) or a hybrid approach based on time series features (T + S or T + P) has proven to yield better results than using spectrogram features (P) alone as input features. When the DNN network directly extracts the original time information features, it comprehensively captures the peak and valley time information. In contrast, the feature extraction process of the STFT spectrogram may result in some loss of accuracy, leading to degraded classification performance. The high dimension of STFT spectral features may also result in low feature correlation. On the other hand, the low dimension of time series data features combined with physiological information features facilitate the capture of pre-correlations and post-correlations between features, thereby improving classification accuracy.

For mental-fatigue-related ECG signal data, the changes in the characteristics of time series data and physiological information data are more critical in the time dimension. Additionally, the relationship between spectral characteristics and time characteristics plays a crucial role.

## 4. Discussion and Conclusions

In this paper, a hybrid algorithm architecture is proposed for the comprehensive processing of ECG signals and the characterization of mental fatigue. In this study, denoised ECG signal data as one-dimensional sequence features, two-dimensional STFT image features, and basic physiological information were integrated into the network to classify mental fatigue. The experimental results show that the accuracy of this design is 95.29%, which has good accuracy for normal ECG signal and mental fatigue data. However, there are still some problems with the study. The experimental dataset is from a non-public dataset and the sample size is still small, which will limit the scope of application of this method.

This method can serve as a dependable parameter to guide dynamic motion schemes during training. By continuously tracking fluctuations in fatigue levels, it can promptly adjust the intensity and quantity of training, thus optimizing the alleviation of exercise-related hazards and enhancing training efficiency. For instance, mental fatigue assessments can guide the gradual reduction in training volume before major competitions. If an athlete is mentally fatigued, reducing cognitive load (e.g., minimizing tactical meetings) may optimize performance while maintaining physical readiness. Similarly, real-time mental fatigue monitoring can help balance training loads during congested competition schedules. For instance, a soccer player recovering from a midweek match might benefit more from light technical drills than intense tactical sessions.

Future research endeavors are expected to center on algorithmic advancements aimed at improving the precision of fatigue assessment, thereby broadening the scope of practical implementation. The ensuing step principally entails devising a means to measure mental fatigue quantitatively, enabling its classification.

## Figures and Tables

**Figure 1 sensors-25-00555-f001:**
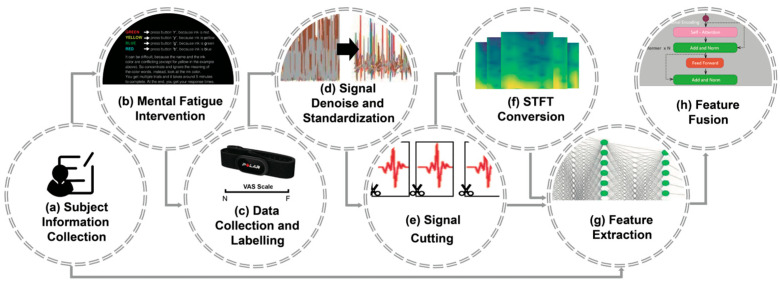
Workflow of our method. Mental fatigue stimulations (**b**) are applied to subjects (**a**). ECG data are then collected and labelled (**c**), followed by denoising and standardization (**d**), and then segmentation (**e**). The preprocessed data are subsequently converted into 2D images using STFT (**f**). Features are extracted from the 1D signal segments, 2D images, and subject-specific information (**g**) using ResNet and Bi-LSTM. Finally, the results are inferred through a transformer by fusing the features (**h**).

**Figure 2 sensors-25-00555-f002:**
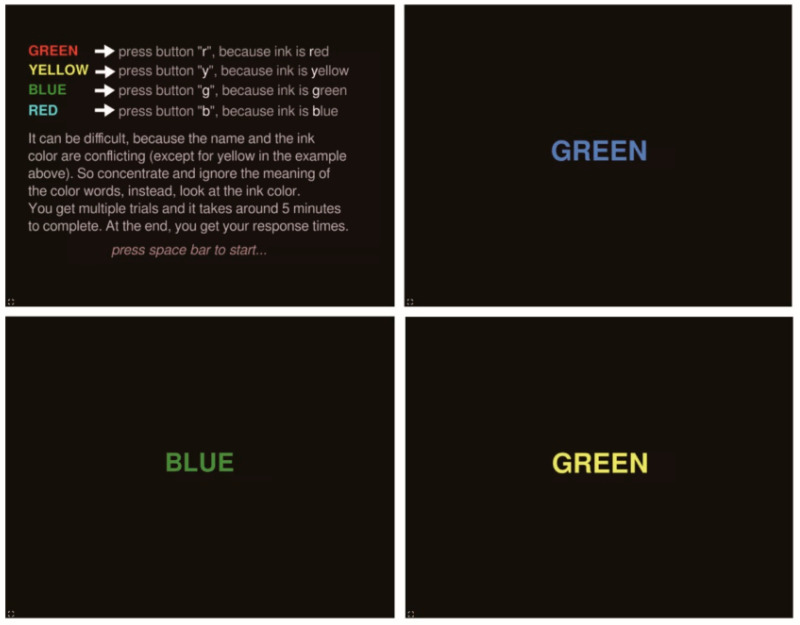
Stroop task description and three example screens.

**Figure 3 sensors-25-00555-f003:**
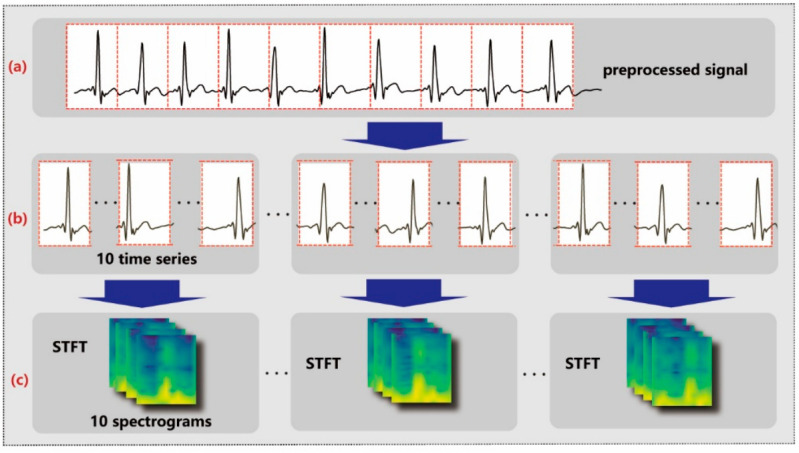
Data preprocessing. The denoised ECG signal (**a**) is segmented into 1D time series (**b**) and converted into 2D spectrograms (**c**).

**Figure 4 sensors-25-00555-f004:**
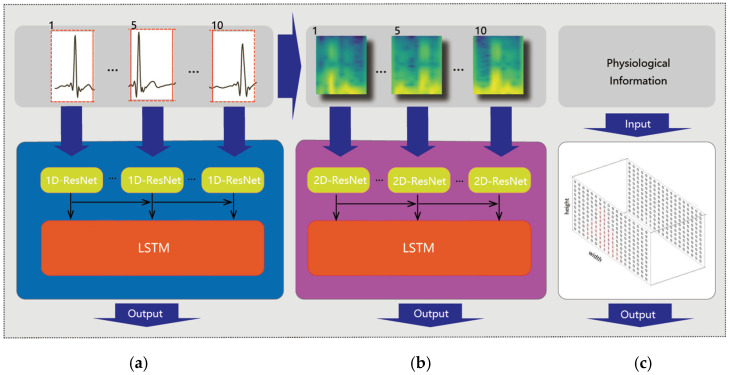
Feature extraction of time series data (**a**), STFT spectrograms (**b**) and physiological information (**c**).

**Figure 5 sensors-25-00555-f005:**
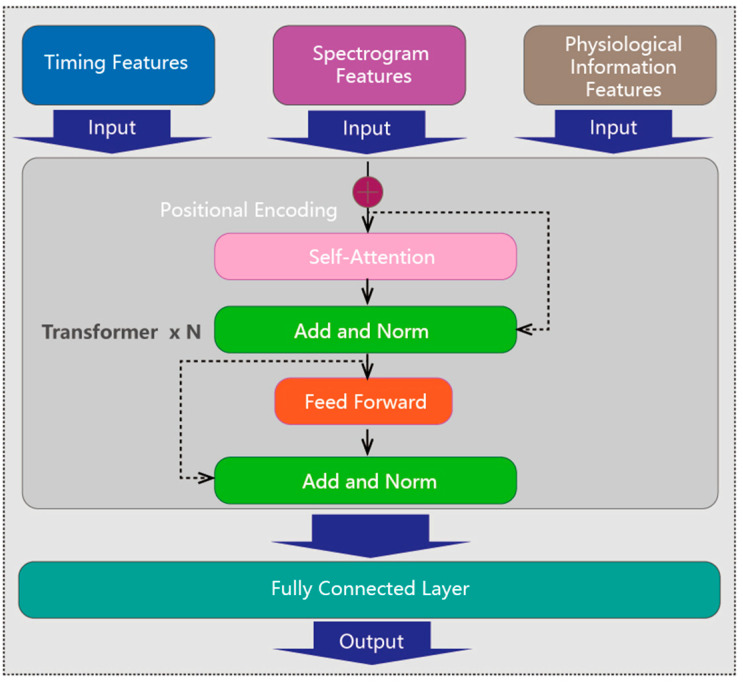
Feature fusion using a transformer encoder.

**Figure 6 sensors-25-00555-f006:**
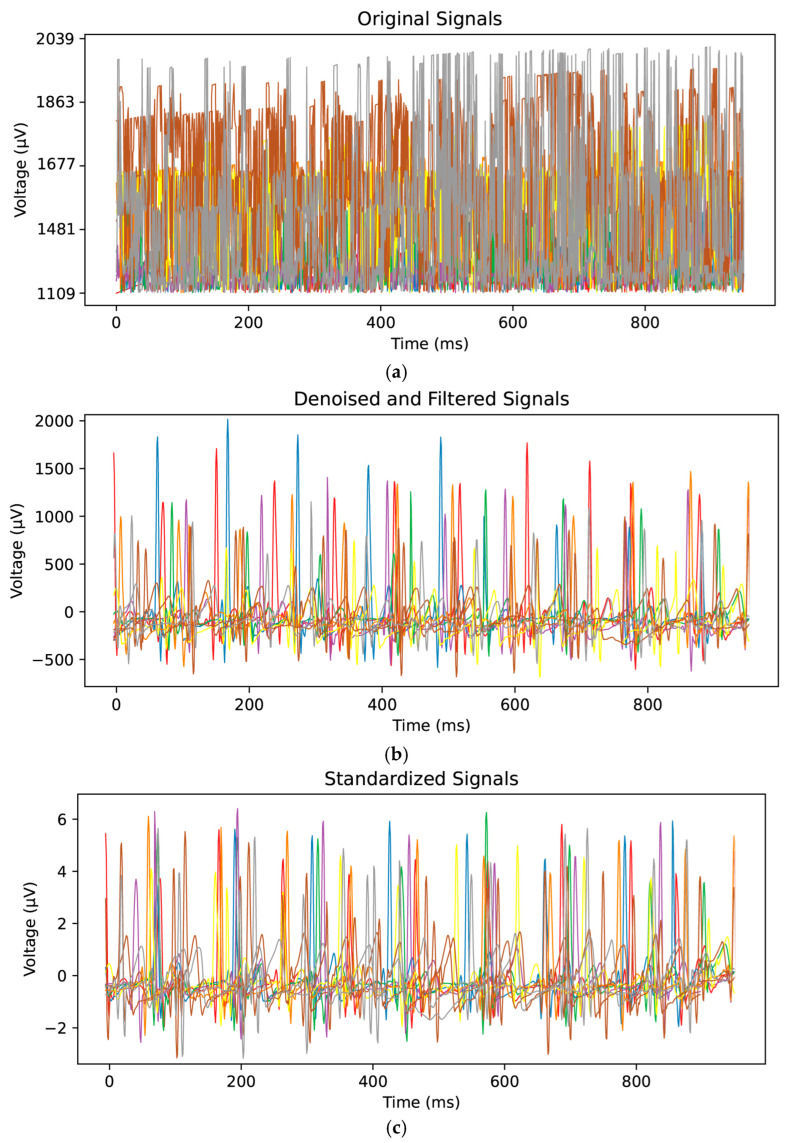
Comparison between the original ECG signal and the preprocessed signal. (**a**) The original ECG signal data, (**b**) ECG signal data after denoising, and (**c**) the baseline-aligned ECG signal after applying the Standard Scaler.

**Figure 7 sensors-25-00555-f007:**
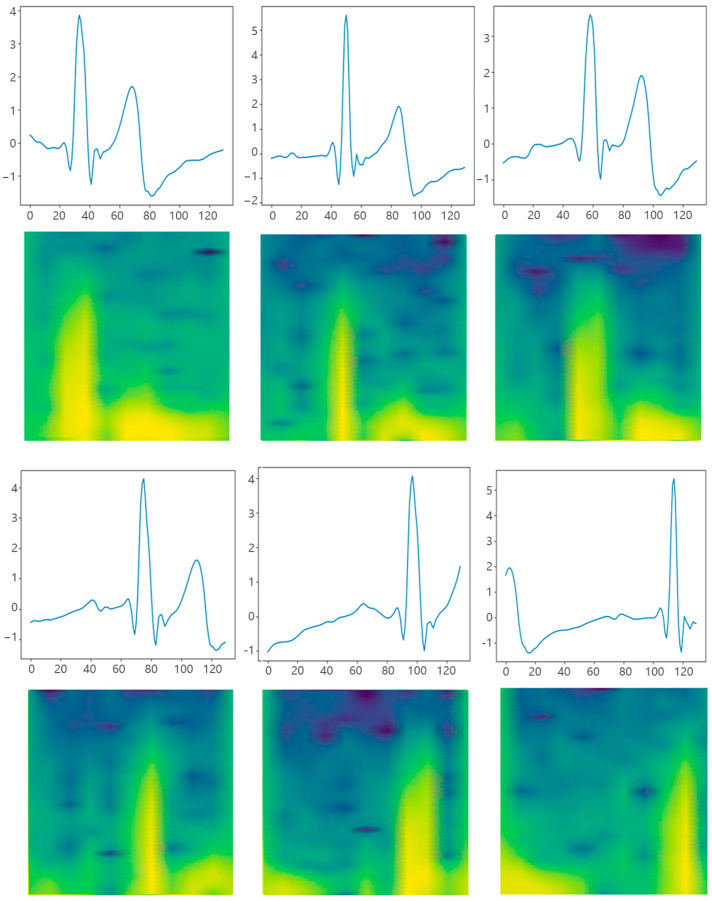
The segmented ECG signal segment is passed through the STFT spectrograms. Each of the figures above displays a time series data segment with 130 points, which represents a 1-second interval.

**Table 1 sensors-25-00555-t001:** Comparison between our algorithm and other algorithms.

	F1	Accuracy (%)
SVM	0.38	55.11
RF	0.57	62.26
CNNs	0.54	69.02
LSTM	0.87	86.67
Bi-LSTM	0.90	90.20
Ours	0.95	95.29

**Table 2 sensors-25-00555-t002:** Ablation study on feature combinations. S means spectrogram features, P means physiological information features, and T means time series features.

Models	F1	Accuracy (%)
S	0.69	69.41
S + P	0.7	71.37
T	0.89	89.41
T + S	0.92	92.54
T + P	0.93	93.33
T + S + P (Ours)	0.95	95.29

## Data Availability

The data presented in this study are available on request from the corresponding author due to the privacy protection of all subjects.

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
