# Peer review of "A Deep Learning Approach for Mental Fatigue State Assessment"

_sensors, 2025, doi:10.3390/s25020555_

Round 1
Reviewer 1 Report
Comments and Suggestions for Authors
1、Introduction: Does this study focus specifically on athletes? If so, it is essential to further emphasize the necessity of assessing mental fatigue in this particular population.
2、Line 89: While discussing the advantages of ECG, this paper highlights its capability for real-time data monitoring and analysis. Has this study successfully achieved real-time monitoring and analysis of mental fatigue?
3、Line 172: Please verify the accuracy of the work flow diagram to ensure consistency with the descriptions of the methodological steps provided in the manuscript.
4、Line 192: Provide an explanation of the validity of using the VAS for assessing mental fatigue.
5、Line 288: Include relevant demographic and occupational information about the participants to give readers a clearer understanding of the study’s sample population.
6、Discussion: The authors state that the proposed method can serve as a reliable parameter to guide dynamic training schemes, allowing for real-time adjustments to training intensity and volume based on fatigue fluctuations. To enhance the practical value of the research, please elaborate on how the findings can be applied in real-world contexts, specifying the implementation process and scenarios.
Comments on the Quality of English LanguageSimplify the long sentences throughout the manuscript to improve clarity and readability.
Author Response
Comment 1: Introduction: Does this study focus specifically on athletes? If so, it is essential to further emphasize the necessity of assessing mental fatigue in this particular population. |
Response 1: Thank you for your valuable advice. We add further discussion on the importance of assessing mental fatigue for athletes in Line 52-54.
|
Comment 2: Line 89: While discussing the advantages of ECG, this paper highlights its capability for real-time data monitoring and analysis. Has this study successfully achieved real-time monitoring and analysis of mental fatigue? |
Response 2: Thank you for your valuable question. We have not developed a fully functional real-time application for assessing mental fatigue yet. But the results from our experiments demonstrate the feasibility of real-time implementation using our method. As outlined in the paper, the ECG signal segment used for an inference of mental fatigue is 10 seconds long. We tested that loading and preprocessing this 10s data costs roughly 1.12 seconds, and the inference costs approximately 0.48s. These timings are already enough for real-time application. Besides, as our code has not been carefully optimized, there is still a large space to enhance performance. For instance, we could leverage CUDA to optimize the preprocessing algorithms, and TensorRT to accelerate the inference. In the future, we are planning to develop this method into a practical, user-friendly application.
|
Comment 3: Line 172: Please verify the accuracy of the workflow diagram to ensure consistency with the descriptions of the methodological steps provided in the manuscript. |
Response 3: Thank you for your valuable advice. We have revised Figure 1 to make our workflow clearer. Besides, we have updated the caption with details and refined the description of the workflow in the main text in Lines 164-183.
|
Comment 4: Line 192: Provide an explanation of the validity of using the VAS for assessing mental fatigue. |
Response 4: Thank you for your valuable advice. We have cited a systematic review on mental fatigue to support the validity and reliability of using VAS in Lines 205-207.
|
Comment 5: Line 288: Include relevant demographic and occupational information about the participants to give readers a clearer understanding of the study’s sample population. |
Response 5: Thank you for your valuable advice. We have added information about the participants in Lines 304-306.
|
Comment 6: Discussion: The authors state that the proposed method can serve as a reliable parameter to guide dynamic training schemes, allowing for real-time adjustments to training intensity and volume based on fatigue fluctuations. To enhance the practical value of the research, please elaborate on how the findings can be applied in real-world contexts, specifying the implementation process and scenarios. |
Response 6: Thank you for your valuable advice. We have added two specific instances on how the findings can be applied in real-world contexts in Lines 458-464.
|

Reviewer 2 Report
Comments and Suggestions for Authors
Please see attachment.

Some of the presentation could be improved to make the results clearer.
Author Response
Comment 1: Please mark the meaning of the horizontal and vertical axes in Fig. 6. |
Response 1: Thank you for your valuable advice. We have marked the meaning of the horizontal and vertical axes in Figure 6 accordingly.
|
Comment 2: Fig. 6 (a) is the original ECG signal data, and Fig. 6 (b) is ECG signal data after denoising. However, the horizontal axes of the two graphs are not the same. Assuming the horizontal axis is time, it is obvious that Fig. 6(b) shows a longer signal. Please explain the process from Fig. 6(a) to Fig. 6(b). It is unreasonable for negative numbers to appear on the vertical axis after noise reduction. |
Response 2: Thank you for your valuable advice. We have revised Figure 6 in accordance with your advice. The range of the horizontal axes and data lengths have been unified in (a), (b) and (c) of Figure 6. Additionally, we have included a description of the process from Figure 6 (a) to (b) in Lines 333-336. The presence of negative numbers is due to the removal of the DC component from ECG signals.
|
Comment 3: The coordinate axes of the EEG signal in Figure 7 are difficult to discern. |
Response 3: Thank you for your valuable feedback. In response to your suggestion, we have enlarged the sub-figures in Figure 7 to enhance clarity. Some of the sub-figures have been deleted as they are merely sample illustrations for STFT spectrograms. The caption of Figure 7 is also updated in Lines 371-373.
|
Comment 4: Table 1 shows the comparison between their proposed algorithm and other algorithms. However, SVM, RF, and CNN have natural disadvantages in this comparison. The SVM and RF were proposed in 1992 and 2001. CNN was used to solve the image problem. In addition, LSTM shows a close F1 score and accuracy to their proposed network. It is considered necessary to compare the capabilities of the network proposed by the authors with Bi-LSTM alone. |
Response 4: Thank you for your valuable feedback. We sincerely apologize that we made an error of recording the LSTM test score in Table 1. We have incorrectly recorded one test result of our model without physiological information (T+S in Table 2) into Table 1, and that’s why the value is very similar to that in Table 2. Additionally, we conducted the Bi-LSTM experiment following your suggestion, and the results have been added to Table 1. The descriptions in the main text have also been updated in Lines 393-402.
|
Comment 5: Fig. 4 shows the feature extraction. It is well known that LSTM is good at dealing with time series problems and ResNet is good at dealing with image or 2D problems. Please explain why the same network structure can handle different types of input. |
Response 5: Thank you for your valuable advice. We have updated Figure 4 and included additional descriptions in Line 259-260 accordingly to make it clear. The convolutional layers in ResNet are 1D layers for time series data, and 2D layers for image input, and the outputs are then fed to LSTM to further extract temporal features. We appreciate your insightful suggestions.
|

Round 2
Reviewer 1 Report
Comments and Suggestions for Authors
1 For Fig.1, what's the meaning of "mental fatigue intervention" of step b? There is no expression about this term in the text. Do you mean "mental fatigue stimulation"?
2 Line 459: "guide tapering", this expression is confusing.
Some expressions need to be improved to be more readable.
For instance: Lines 458-463.
Author Response
Comment 1: For Fig.1, what's the meaning of "mental fatigue intervention" of step b? There is no expression about this term in the text. Do you mean "mental fatigue stimulation"? |
Response 1: Thank you for your valuable advice. The intervention of mental fatigue is indeed the stimulation to induce mental fatigue. We have changed the word to stimulation in Line 180 according to your advice.
|
Comment 2: Line 459: "guide tapering", this expression is confusing. |
Response 2: Thank you for your valuable question. We have clarified the word “tapering” in Line 461 by a more specific description. |

Reviewer 2 Report
Comments and Suggestions for Authors
The authors' revisions meet my requirements.
In addition, I still feel puzzled by the changes in the vertical axes of the three graphs in Figure 6.
Author Response
Comment 1: The authors' revisions meet my requirements. In addition, I still feel puzzled by the changes in the vertical axes of the three graphs in Figure 6. |
Response 1: Thank you for your valuable advice. We have tried to make the description of changes in the vertical axes clearer in Lines 337-339. |
